# Monitoring Opioid-Use-Disorder Treatment Adherence Using Smartwatch Gesture Recognition

**DOI:** 10.3390/s25082443

**Published:** 2025-04-12

**Authors:** Andrew Smith, Kuba Jerzmanowski, Phyllis Raynor, Cynthia F. Corbett, Homayoun Valafar

**Affiliations:** 1Department of Computer Science and Engineering, University of South Carolina, Columbia, SC 29201, USA; andrewsmith@sc.edu (A.S.); jakubj@email.sc.edu (K.J.); 2Advancing Chronic Care Outcomes Through Research and iNnovation (ACORN) Center, Department of Biobehavioral Health & Nursing Science, College of Nursing, University of South Carolina, Columbia, SC 29201, USA; raynorp@mailbox.sc.edu (P.R.); corbett@sc.edu (C.F.C.)

**Keywords:** machine learning, neural networks, human activity recognition, smart healthcare, ecological momentary assessment, context-aware environments, medication detection, wearable sensors

## Abstract

The opioid epidemic in the United States has significantly impacted pregnant women with opioid use disorder (OUD), leading to increased health and social complications. This study explores the feasibility of using machine learning algorithms with consumer-grade smartwatches to identify medication-taking gestures. The research specifically focuses on treatments for OUD, investigating methadone and buprenorphine taking gestures. Participants (n = 16, all female university students) simulated medication-taking gestures in a controlled lab environment over two weeks, with data collected via Ticwatch E and E3 smartwatches running custom ASPIRE software. The study employed a RegNet-style 1D ResNet model to analyze gesture data, achieving high performance in three classification scenarios: distinguishing between medication types, separating medication gestures from daily activities, and detecting any medication-taking gesture. The model’s overall F1 scores were 0.89, 0.88, and 0.96 for each scenario, respectively. These findings suggest that smartwatch-based gesture recognition could enhance real-time monitoring and adherence to medication regimens for OUD treatment. Limitations include the use of simulated gestures and a small, homogeneous participant pool, warranting further real-world validation. This approach has the potential to improve patient outcomes and management strategies.

## 1. Introduction

Currently, in the United States (US), the opioid epidemic continues to escalate, with over 10 million people misusing opioids and over 80,000 deaths attributed to drug overdose annually [1]. An exceptionally vulnerable and underresourced subset of people who misuse opioids are pregnant women with opioid use disorder (OUD). Over 50,000 pregnant people have opioid use disorders in the US, which imposes an additional USD 462 million annually on the Medicaid system in the United States alone [2,3]. Drug overdose deaths among pregnant and postpartum people nearly doubled between 2017 and 2020 [4,5], largely attributed to an increase in illicitly manufactured synthetic opioids (fentanyl) since the onset of the COVID-19 pandemic [6,7,8]. In addition to the risk of overdose deaths, pregnant people with OUD are at risk for many social, emotional, and health complications that include stigma, intimate partner violence, trauma, sexually transmitted diseases, and cardiac events [6,9,10,11]. Compared to pregnant people without OUD, pregnant people with OUD tend to receive less prenatal care [9], and their infants are at risk for premature birth, low birth weight, neonatal opioid withdrawal syndrome (NOWS), congenital malformations, and infant and childhood neurodevelopmental delays and problems [9,12,13,14,15].

There is a clear need to develop innovative strategies that support treatment adherence and long-term recovery for pregnant people with OUD. Medications for opioid use disorder (MOUD) are evidence-based treatments for individuals with OUD [10,11]. Synthetic prescription medications, full opioid agonists (e.g., methadone), and partial opioid agonists (e.g., buprenorphine), have been developed to reduce opioid cravings, withdrawal symptoms, and the body’s response to future drug use [10,11]. Despite strong evidence of the effectiveness of MOUD, they are significantly underutilized among the general population [16,17,18,19] and pregnant people specifically [20]. Additionally, there are significant challenges in retaining pregnant people in MOUD treatment once the baby has been delivered. Digital health interventions have been increasingly used to support treatment adherence and self-management for people with SUD [21,22,23,24], with even greater utilization occurring during and after the COVID-19 pandemic [21,22,23,24,25]. Pregnant people with OUD, may struggle to track which medications they have taken and when, especially if their condition impairs them. This lack of information hinders clinicians’ ability to optimize MOUD treatment.

Existing adherence monitoring methods, such as smart pill bottles and vision-based techniques, provide only partial solutions [26,27]. Smart pill bottles can be deceived and do not confirm ingestion [27]. Vision-based techniques have also been explored, but are limited by lower accuracy, expensive equipment, and a lack of mobility, making them impractical for real-world adherence monitoring [27]. More modern techniques, such as earbuds that detect swallowing, have shown promising results in confirming ingestion; however, they require continuous wear, which may not be practical for most individuals [26]. While wrist-worn wearable devices are not perfect, they offer unique advantages over other methods. Unlike pill bottles, they provide a better ability to determine whether an individual has taken their medication, though they may not be as precise as earbuds. Compared to earbuds, they are less likely to be lost during daily activities and do not carry the same social stigma, as many workplaces prohibit earbuds but not smartwatches. Additionally, many medications do not come in pill bottles, further limiting the effectiveness of pill-based adherence solutions. With advances in deep learning [28] and the increasing ubiquity of wearable health devices—44% of Americans now own a wearable health-tracking device [29]—these technologies present an opportunity to monitor and engage patients in MOUD adherence in real time. Moreover, integrating wearable technologies with existing health tools, such as smart health cards, could further enhance adherence tracking by consolidating medication data alongside other critical patient information, such as the patient’s trimester. Presenting this information in a secure and standardized manner to healthcare providers could improve treatment optimization [30]. To that end, this research aims to establish the feasibility of identifying medication-taking gestures for opioid use disorder using consumer-grade smartwatches and machine learning algorithms.

Advances in deep learning [28] and wearable health devices like smartwatches and Fitbits and their growing ubiquity—44% of Americans now own a wearable health-tracking device [29]—present an opportunity to monitor and engage patients in taking MOUD in real-time. If successful, engaging patients in their care using wearable devices will help better adjust MOUD based on symptoms and optimize patient outcomes. To that end, this research aims to establish the feasibility of identifying medication-taking-gestures for opioid use disorder using machine learning algorithms and consumer-grade smartwatches.

The rest of this paper is structured as follows: Section 2 details the methodology, beginning with the data-collection protocol and data labeling process. This is followed by data-preprocessing steps, including segmentation, standardization, and data splitting into training, validation, and test sets. The model architecture, training procedure, and hyperparameter selection are then described. The validation paradigm and evaluation metrics define how model performance was assessed. Section 3 presents experimental results across three classification scenarios, while Section 4 discusses the findings, their implications, and future research directions.

## 2. Materials and Methods

Medication to treat OUD commonly comes in two forms: both are consumed orally; one, methadone, is liquid and is dispensed from a small, single-dose bottle. The other, buprenorphine, is dispensed either as a pill or in the form of a strip, similar to Listerine, and is taken sublingually by placing either the pill or the strip under the tongue. As a proof-of-principle, we embarked on simulating medication-taking gestures, allowing for controlled, large-scale data collection. To assess the feasibility of identifying OUD medication gestures using machine learning algorithms and consumer-grade smartwatches, we collected data in the following three categories of activities:Simulated methadone-taking gesture;Simulated buprenorphine-taking gesture in sublingual strip form;Performing typical daily gestures unrelated to medication-taking.

In an ideal scenario, we aim to detect and distinguish between the micro-gestures associated with taking buprenorphine, methadone, and other daily gestures. Some inherent limitations exist in our proposed approach. For example, consuming water from a small bottle produces a gesture identical to consuming methadone from a similar bottle. However, additional factors can help differentiate these actions. These include heart rate, blood oxygen levels, time of day, and GPS data. All of these metrics are obtainable from consumer-grade smartwatches.

This similarity in gestures does, however, pose an advantage at this stage of development. Due to this overlap, we can simulate medication-taking gestures without actually using real medication, which allows for easier and larger-scale data collection. To assess the feasibility of identifying OUD medication gestures using machine-learning algorithms and consumer-grade smartwatches, we propose the following experiment:Data Collection: Sensor data was collected from individuals performing:(a)Simulated methadone (liquid ingestion).(b)Simulated buprenorphine (sublingual film ingestion).(c)Daily living activities unrelated to medication-taking.Data Processing: Raw sensor data were filtered, segmented into fixed-length windows, and labeled for supervised learning.Model Training and Evaluation: Data were split into training, validation, and test sets. Models were trained to classify gestures, optimized using validation data, and evaluated on the test set using F1 score, precision, recall, and confusion matrices.Representation Analysis: Gesture embeddings were visualized using t-SNE to assess learned feature separability.Deployment Considerations: Model performance on unseen participants and feasibility for real-time smartwatch classification were assessed.

### 2.1. Participants

Sixteen adults from a student population at a significant university in the southeastern U.S. were selected for this research. Each was given a detailed briefing on the study’s aims, methods, potential risks, and benefits prior to their participation. After this, all participants willingly provided verbal informed consent, following the ethical guidelines approved by the university’s institutional review board (IRB). They were also informed of their right to withdraw from the study at any time without facing any repercussions.

The participant demographics included an average age of 22.8 years (spanning from 19 to 55 years), with all participants identifying as female (100%). The ethnic makeup was predominantly White at 93%, with 6% African American and another 6% Asian participants; note that the total exceeds 100% due to some individuals identifying with multiple ethnicities. Handedness was also noted, with 81% of the participants being right-handed and 19% left-handed.

To maintain participant confidentiality and comply with ethical standards, all data were anonymized. Personal identifiers were kept secure and only accessible to authorized study personnel, ensuring adherence to data protection laws. This methodical approach to participant selection and data management was crafted to ensure an ethical, respectful, and scientifically valid research setting.

### 2.2. Equipment

This study employed two models of Android smartwatches, the Ticwatch E and Ticwatch E3, both running on Google Wear OS. These devices were selected due to their strong capabilities in real-time data processing and their compatibility with our tailored software. We utilized a custom application, ASPIRE v4.0 [31], which was specifically engineered for Android smartwatches. This app was instrumental in gathering sensor data and was crucial for the detection and analysis of human gestures. The software configuration allowed for continuous data collection at a sampling rate of 100 Hz per channel, ideal for capturing the subtle movements associated with activities like eating, performing yoga-like poses, taking medication, and simulating smoking, as described in prior research [32]. This approach was favored for its straightforwardness in managing the large datasets produced at high frequencies, though it increased the demand for data-storage capacity. The storage issue was manageable in this study, but for future experiments requiring over 13 weeks of data recording, we plan to implement binary data recording. The combination of advanced wearable technology, custom software, and stringent data-management protocols was meticulously designed to ensure the collection of high-quality sensor data. These steps are essential for the precise analysis of gesture-based activities and play a pivotal role in the success of this study in advancing human activity recognition through wearable technology. The ASPIRE software can be requested to aid further research in this domain.

### 2.3. Data Collection and Storage

We established a structured protocol where participants engaged in predefined activities in a controlled laboratory environment. Participants attended an initial training session where, following informed consent, they received smartwatches and instructions for simulating medication-taking events (MTEs) (Figure A1 and Figure A2) during structured data collection. All data collection occurred in the laboratory or during live video-recorded meetings, ensuring that participants were supervised while performing the required gestures. The data collection was organized into two phases described below:Natural MTE (nMTE): Participants performed medication-taking gestures naturally for one week (5 days).Scripted MTE (sMTE): Participants performed gestures following a scripted protocol for a second week (5 days).

Participants wore the smartwatch which includes triaxial sensors. The smartwatch captured acceleration and gyroscope data in the x, y, and z axes, along with timestamps, only during these supervised sessions, capturing movement exclusively during the simulation of medication-taking events, and did not continuously monitor participants. During each data-collection session, participants performed 20 gestures in total: 10 gestures for methadone simulation and 10 for buprenorphine simulation. For each type of medication, participants performed five repetitions wearing the watch on their left hand and five on their right hand. Regardless of which wrist wore the watch, participants executed the gestures according to their current study phase—nMTE or sMTE. The data were sampled at a rate of 100 Hz. Sensor data from each participant were recorded and stored directly on the smartwatch in CSV format.

Participants received a USD 25 gift card as an incentive after collecting and returning 10 days of data using the watch. Once the watch was returned, the CSV files were securely transferred to a Linux-based server for further processing, analysis, model development, and model validation. The raw data wERE then uploaded and stored in Google cloud. Raw data included timestamps (hour, minute, second, and millisecond), dates (day, month, and year), and accelerometer or gyroscope readings for the x, y, and z axes.

### 2.4. Data Labeling

To ensure accurate, high-quality labeling, participants were instructed to vigorously shake their wrists before and after each MTE. Video recordings further assisted in accurately identifying an action and labeling the MTE. Research team members who served as ML supervisors visually confirmed MTE gestures to ensure data quality. To facilitate a streamlined labeling process a custom utility program was developed and used to generate a final dataset ready for ML training and testing.

Figure 1 and Figure 2 highlight clear differences in accelerometer readings (x, y, z axes) when comparing gestures for sublingual films (Listerine strips) versus oral liquid (water). Insufficient confidence by supervisors in accurately labeling some gestures, as well as some participants not performing all 20 gestures every session, resulted in a final dataset containing 1409 oral liquid methadone (water) gestures and 1331 sublingual buprenorphine (Listerine) gestures. After labeling and processing, the dataset was prepared for use in the AI/ML model.

### 2.5. Data Preprocessing

To ensure robust model development and evaluation, data were preprocessed and split. Gesture recordings varied in duration from 2.5 s to 11 s (Figure 3 shows the distribution of gesture durations for both types of gestures). Thus, to address the fixed-length input requirement of neural networks, we implemented a windowing approach. A 4-s window length was selected, as it preserved 96% of liquid oral methadone (water) gestures and 95% of sublingual buprenorphine (Listerine) gesture recordings, maximizing data retention while maintaining a standardized input format. In this study, data were split at the gesture level such that no individual gesture was in both training and testing sets. The final dataset was divided into training (80%) and testing (20%) sets, with stratification by gesture class to ensure balanced representation. Additionally, a small proportion of the training set was split into a validation set for model selection (early stopping, hyperparameter tuning, etc). The gesture-level splitting ensured that:

To achieve statistically meaningful results, training and testing sets must be both independent and identically distributed. However, defining independence in human activity data, such as gesture recognition, can vary depending on the granularity of the split.

### 2.6. Independence Considerations

We considered multiple levels of independence to design the data split:Participant Level: Ensuring that no data from the same participant appears in both training and testing sets would represent the most stringent definition of independence. However, this assumption may be overly strict if the system is intended for deployment within the same group of participants it was trained on.Recording Level: Each participant provided multiple recordings, captured at different times, which may exhibit variations (e.g., differences in how the smartwatch was worn). Splitting recordings into separate training and testing sets would partially enforce independence but may still allow for intra-participant correlations.Gesture Level: Within each recording, gestures often exhibit self-similarity due to consistent movement patterns. Splitting at the gesture level ensures that no individual gesture contributes samples to both training and testing sets, minimizing overlap and ensuring independence.

Given these considerations, we adopted a gesture-level splitting strategy. This approach ensures that no data samples overlap between training and testing sets, as each sample corresponds to an independent bout of activity. While this strategy may still allow data from the same participant or recording to appear in both sets, it balances the need for independence with practical considerations for deployment scenarios.

### 2.7. Model Design

Human activity data encompass both spatial and temporal characteristics. To facilitate the automatic extraction of feature-rich information for subsequent classification, we employed a convolutional neural network (CNN). Residual Networks (ResNets) [33] are renowned for their ability to manage spatial hierarchies and are often equated with CNNs, primarily due to the efficacy of residual connections. Our model specifically adopts a RegNet-style [34] 1D ResNet, which offers a more constrained and organized hyperparameter design space, resulting in superior performance. This model was chosen over other popular architectures for its relative simplicity, the unique ability to recognize hierarchical spatial characteristics, ease of training, and for not being the most data-hungry.

The model accepts 3-channel, 100 Hz raw accelerometer signals as input, which it then transforms into a high-dimensional feature space via the RegNet-style ResNet architecture.

For all components of the model, training and evaluation were performed using only the dataset described above, with the recordings split into training and testing subsets.

### 2.8. Optimization and Hardware Utilization

The models were optimized using the AdamW [35] optimizer, which is an enhancement of the conventional Adam [36] optimizer by integrating a decoupled weight decay regularization. This form of weight decay contributes to better training stability and convergence. Gradient descent updates were effectively distributed across two NVIDIA GeForce RTX 4090 GPUs, leveraging PyTorch 2.5.1 [37]. The use of these high-performance GPUs facilitated the management of the computational load associated with training deep neural networks, accelerating the training process and boosting model performance via parallel processing. This approach not only ensures efficient learning but also maximizes the model’s ability to generalize to new, unseen data, which is vital for real-world applications where diverse human activities must be accurately identified and understood.

### 2.9. Evaluation Scenarios

To comprehensively evaluate the model’s performance in classifying gestures, we designed three distinct evaluation scenarios. Each scenario captured a different level of complexity and served a unique purpose in assessing the model’s applicability to real-world use cases.

#### 2.9.1. Scenario 1

The first scenario focused on distinguishing between sublingual films and oral liquid gestures. These gestures, while differing in intent, share similar movement patterns, making them difficult to distinguish. This two-class scenario was designed to test the model’s ability to capture subtle variations in gesture dynamics. Such differentiation is valuable in clinical settings where distinguishing between medication types is crucial for adherence monitoring and intervention strategies. Performance metrics, including precision, recall, F1 score, and confusion matrices, were used to evaluate the model in this scenario.

#### 2.9.2. Scenario 2

In the second scenario, the model was tasked with classifying gestures into three classes: sublingual films, oral liquids, and daily living gestures. This scenario reflects a more realistic environment where medication-taking gestures occur alongside unrelated daily activities. The addition of daily gestures introduces increased complexity, as the model must simultaneously distinguish between inter-medication differences and unrelated movements. Metrics such as macro-averaged precision, recall, and F1 score were used to assess the model’s performance across all classes, while confusion matrices provided insights into class-specific misclassifications.

#### 2.9.3. Scenario 3

The third scenario simplified the task by combining sublingual and oral liquid gestures into a single “medication-taking” class and contrasting it with daily living gestures. This two-class scenario prioritizes detecting medication-taking gestures over distinguishing specific medication types. It represents a practical approach to monitoring adherence, where the primary goal is to identify whether a gesture involves medication-taking behavior. The model’s performance in this binary classification task was evaluated using precision, recall, and F1 score for the medication-taking class, with confusion matrices highlighting false positives and false negatives.

These three scenarios provide a comprehensive framework to assess the model’s capabilities, ranging from fine-grained classification of medications to broader, more practical adherence detection. The results of these evaluations, detailed in the Section 3, offer insight into the trade-offs between complexity and performance.

### 2.10. Evaluation Metrics

Defining metrics for evaluating deep learning models is essential to ensure valid results. Cross-entropy loss serves as a robust metric for classification [38,39,40,41]; however, it presents challenges in interpretation. A frequent pitfall in evaluating imbalanced classification scenarios is relying on accuracy as a performance indicator. Deep learning models can easily learn to exploit class imbalance via mini-batches, leading to biased predictions. To avoid artificially inflated accuracy due to such biases, we set out to maximize two objectives. For activity recognition, the first goal is to accurately predict as many true activities as possible. The second is to reduce the number of instances in which activities are incorrectly identified. These objectives align with the metrics of precision and recall.

Precision and recall are derived from a confusion matrix. To merge these into a single reportable metric, we introduce the F1 score, which is the harmonic mean of precision and recall, with values ranging from 0 to 1, similar to accuracy, where 1 indicates a perfect classifier [42].

Each of these metrics plays a crucial role in offering a comprehensive view of the performance of the model, identifying its strengths and weaknesses, and thus steering further development and deployment in practical scenarios.

## 3. Results

### 3.1. Dataset Qualitative Results

The data-collection process produced approximately 10 recordings for each of the 16 participants. Sensor data were sampled at 100 Hz with each recording having a timestamp and a 3-dimensional acceleration vector for every data point. On average, participants contributed 1.2 h of data, corresponding to roughly 432,000 samples per individual. In total, this resulted in approximately 6.912 million data points, representing 19.2 h of human activity data collected at a 100 Hz sampling rate.

### 3.2. Scenario 1: Binary Classification of Medication Types

In Scenario 1, the model was tasked with differentiating between two medication-taking gestures: methadone (liquid) and buprenorphine (sublingual film). The training and validation loss curves (Figure 4A) demonstrate stable convergence, with validation loss plateauing after approximately 10 epochs. The F1 score curves (Figure 4B) indicate balanced classification performance across both medication types.

The model achieved a macro-averaged F1 score of 0.89, with precision and recall values of 0.89 and 0.88, respectively (Table 1). Performance was slightly higher for methadone gestures (F1: 0.91, with recall: 0.92 and precision: 0.90) compared to buprenorphine (F1: 0.86, with recall: 0.84 and precision: 0.88). The precision and recall matrices (Figure 4D,E) highlight that the model correctly classified most gestures, though some misclassifications occurred due to overlapping motion patterns.

The t-SNE visualization (Figure 4F) shows that methadone and buprenorphine gestures formed distinguishable but closely related clusters. The t-SNE plots reveal that methadone and buprenorphine gestures cluster closely, indicating a high degree of similarity in their motion patterns, while a few outliers, likely due to individual variations in gesture execution, highlight the need for personalized model adjustments. The model performs slightly better for methadone gestures likely due to their greater distinctiveness from daily activities, as evidenced by clearer separation in the feature space and fewer misclassifications in the confusion matrix. These results suggest that while the model effectively captures gesture differences, the similarity between liquid and film ingestion leads to occasional misclassifications.

### 3.3. Scenario 2: Three-Class Classification (Including Daily Living Gestures)

In Scenario 2, a third class—daily living gestures—was introduced to simulate a more naturalistic setting. As shown in the training and validation loss curves (Figure 5A), the addition of this third category increased training complexity, with validation loss stabilizing after approximately 12 epochs. The F1 score curves (Figure 5B) indicate a moderate decline in performance compared to Scenario 1, particularly in distinguishing methadone and buprenorphine gestures from daily living gestures.

The model obtained a macro-averaged F1 score of 0.88, with precision and recall values of 0.88 and 0.87 (Table 1). Methadone gestures were classified with an F1 score of 0.87 (recall: 0.85, precision: 0.90), while buprenorphine gestures had an F1 score of 0.80 (recall: 0.79, precision: 0.81). The daily living class was identified with high confidence (F1 score: 0.96, recall: 0.99, precision: 0.94), suggesting that adding a non-medication category improved the model’s ability to distinguish medication-taking from unrelated activities.

The confusion matrices (Figure 5D,E) reveal that the most frequent misclassifications occurred between methadone and buprenorphine, similar to Scenario 1. However, the model consistently classified daily living gestures correctly. The t-SNE visualization (Figure 5F) supports this observation, with daily living gestures forming a distinct cluster separate from medication-taking gestures.

### 3.4. Scenario 3: Binary Classification of Medication-Taking vs. Daily Living

Scenario 3 simplified the classification task by grouping both methadone and buprenorphine gestures into a single “medication-taking” class, contrasted against daily living gestures. The loss curves (Figure 6A) show faster convergence than in Scenario 2, reflecting the reduced complexity of a binary classification task. The F1 score curves (Figure 6B) indicate that the model consistently improved performance with each epoch.

The model achieved its highest overall accuracy in this scenario, with a macro-averaged F1 score of 0.96, and precision and recall values of 0.96 (Table 1). The medication-taking class was classified with an F1 score of 0.96 (recall: 0.97, precision: 0.95), while the daily living class achieved similar performance (F1 score: 0.96, recall: 0.94, precision: 0.97).

The confusion matrices (Figure 6D,E) confirm that this binary classification approach resulted in minimal false positives and false negatives, showing strong separation between medication-taking and daily living activities. The t-SNE visualization (Figure 6F) further highlights the effectiveness of this approach, with two well-separated clusters representing the two classes. These results suggest that a binary classification model may be optimal for real-world applications where detecting medication intake is the primary goal.

## 4. Conclusions

This study demonstrates the feasibility of utilizing consumer-grade smartwatches and machine learning algorithms to detect medication-taking gestures in individuals with opioid use disorder (OUD). Through the collection and analysis of gesture data, our model was able to distinguish between simulated methadone and buprenorphine intake as well as differentiate these from daily living activities. The results indicate that deep learning models, particularly convolutional neural networks (CNNs) with a RegNet-style ResNet architecture, can successfully classify medication-taking behaviors with high precision and recall. These findings highlight the potential for wearable technology to enhance medication adherence monitoring in OUD treatment, paving the way for more personalized, optimized, and data-driven interventions.

## 5. Discussion

The results from our three evaluation scenarios underscore the effectiveness of our approach in detecting and classifying medication-taking gestures. Notably, the highest performance was observed in Scenario 3, where the model achieved an F1 score of 0.96 in distinguishing medication-taking from daily living gestures. This suggests that, while differentiating between specific medication forms presents challenges due to gesture similarity, the ability to broadly identify medication adherence behavior remains highly reliable.

One of the key advantages of this approach is its potential to provide real-time adherence monitoring without relying on self-reported data, which is often subject to bias and inaccuracies. Wearable health-tracking devices, which are increasingly adopted by the general population, offer a non-invasive and scalable solution for supporting individuals undergoing medication for opioid use disorder. The integration of additional smartwatch features, such as heart-rate monitoring, blood oxygen levels, and GPS tracking, could further improve accuracy and contextual awareness. The model can integrate with healthcare systems by securely transmitting adherence data to electronic health records (EHRs) and enabling real-time alerts for providers, enhancing care coordination for patients with opioid use disorder (OUD). Ethically, the continuous monitoring of pregnant women requires informed consent, data-privacy safeguards, and a focus on support rather than surveillance to avoid stigmatization.

Despite the promising results, several limitations must be acknowledged. First, the study relied on simulated medication-taking gestures rather than real-world ingestion of methadone or buprenorphine, which may introduce discrepancies when deployed in practical settings. Future research should aim to validate these findings in clinical or real-life environments to assess the robustness of the model in detecting actual medication adherence. Although the model achieved a high F1 score of 0.96 in distinguishing medication-taking gestures from daily activities, its performance in real-world clinical settings remains to be validated. Future research will focus on testing the model in diverse, uncontrolled environments to ensure its reliability and minimize the risk of false positives and negatives in high-stakes applications. Second, while the dataset was sizable, it was collected from a relatively homogeneous participant pool, predominantly young female individuals. Expanding the study to a more diverse population would enhance the generalizability of the findings. Third, while the RegNet-style 1D ResNet demonstrated strong performance in this feasibility study (F1 scores of 0.89, 0.88, and 0.96), the lack of comparison with other deep learning architectures, such as LSTMs or Transformer-based models, limits our ability to assess its relative efficacy; future work will include systematic benchmarking to optimize model selection for real-world deployment. In addition, in subsequent studies, we plan to enhance the interpretability of our model by integrating advanced explainable AI methods, such as SHAP and Grad-CAM, to provide deeper insight into its decision-making processes. While the study utilized Ticwatch E and E3 smartwatches, future work will test the model across various smartwatch models to ensure cross-device robustness. Participants were instructed to wear the watch consistently, but real-world variations in wearing style may impact performance; future studies will explore this variability.

Another consideration is the potential for false positives and negatives, particularly when distinguishing between medication-taking and similar daily activities. Refinements in sensor fusion techniques, such as incorporating gyroscope data with contextual metadata, could improve differentiation and minimize misclassifications. Additionally, leveraging advancements in self-supervised learning may enhance the model’s ability to generalize across various users and environments with minimal manual data labeling.

## 6. Future Directions

To further advance this research, several key steps are recommended:Clinical Validation: Conduct real-world studies with individuals actively undergoing OUD treatment to assess the practical effectiveness of the model.Feature Expansion: Incorporate additional smartwatch sensors, such as pulse oximetry, heart rate, time of day, and temperature monitoring, to improve classification performance.User Adaptability: Develop personalized machine learning models that can adjust to individual variations in gesture performance over time.Integration with Treatment Programs: Explore integration with mobile health applications and digital therapeutic platforms to provide real-time feedback and adherence support.

In conclusion, this study provides a foundational step toward leveraging wearable technology and AI-driven analysis for medication adherence monitoring in OUD treatment. While challenges remain, the potential for improving patient outcomes through real-time, automated monitoring is significant, warranting further exploration and development.

## Figures and Tables

**Figure 1 sensors-25-02443-f001:**
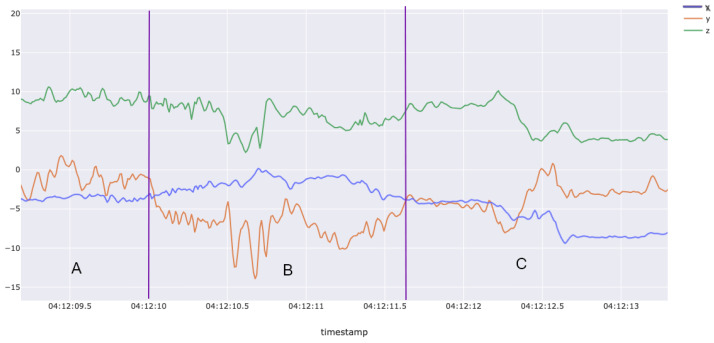
Example smartwatch accelerometer data for the buprenorphine gesture. This figure illustrates accelerometer data collected during a simulated buprenorphine-taking gesture. The motion pattern highlights distinct sub-actions: (**A**) tearing open the top of the packet and bringing the medication toward the mouth; (**B**) placing the sublingual film under the tongue; and (**C**) lowering the hand back down.

**Figure 2 sensors-25-02443-f002:**
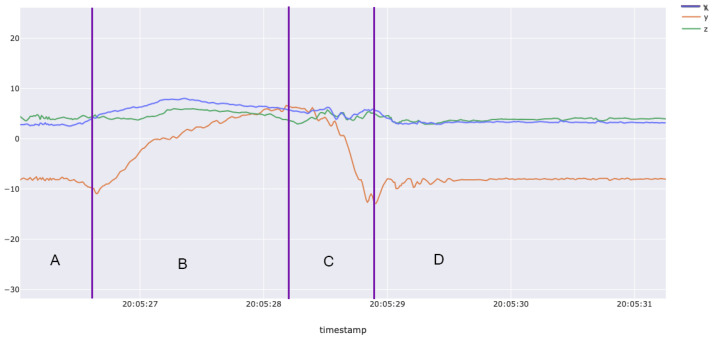
Example smartwatch accelerometer data for the methadone gesture. This figure presents accelerometer recordings from a smartwatch worn during a simulated methadone-taking gesture. The motion trajectory illustrates key phases of the action: (**A**) unscrewing the cap (not visible in accelerometer data, as the hand without the watch performed this action); (**B**) raising the bottle to the mouth and ingesting the liquid; (**C**) lowering the bottle back down; and (**D**) the completion of the gesture.

**Figure 3 sensors-25-02443-f003:**
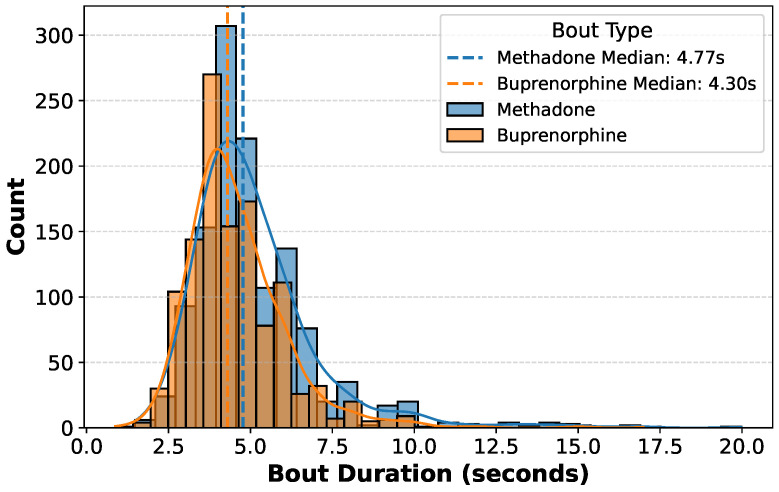
The distribution of bout durations for methadone and buprenorphine. The histogram represents the frequency of bout durations (in seconds), with overlaid kernel density estimates (KDEs) for both substances. Dashed vertical lines indicate the median bout duration for each group. Methadone is shown in blue, while buprenorphine is shown in orange.

**Figure 4 sensors-25-02443-f004:**
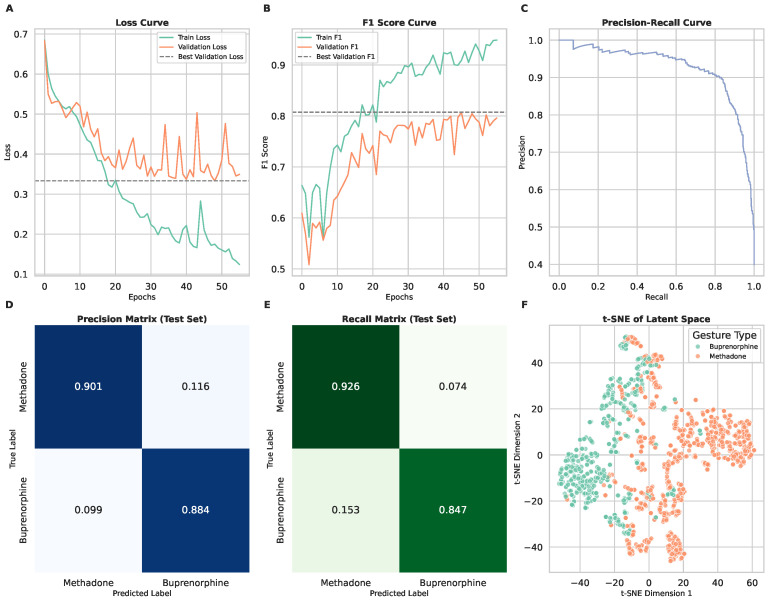
Model performance and gesture classification results for Scenario 1. In this scenario, the model performs binary classification to differentiate between two medication-taking gestures: methadone (liquid) and buprenorphine (sublingual film). (**A**) Training and validation loss curves, showing model convergence and generalization over epochs. (**B**) F1 score curves for training and validation sets, indicating classification performance across medication types. (**C**) Precision–recall curve demonstrating the model’s ability to distinguish between methadone and buprenorphine gestures. (**D**) Precision matrix of the test set, illustrating the proportion of correctly classified gestures for each medication type. (**E**) Recall matrix of the test set, highlighting model sensitivity to each medication-taking gesture. (**F**) t-SNE visualization of latent space representations, showing the clustering of methadone and vuprenorphine gestures.

**Figure 5 sensors-25-02443-f005:**
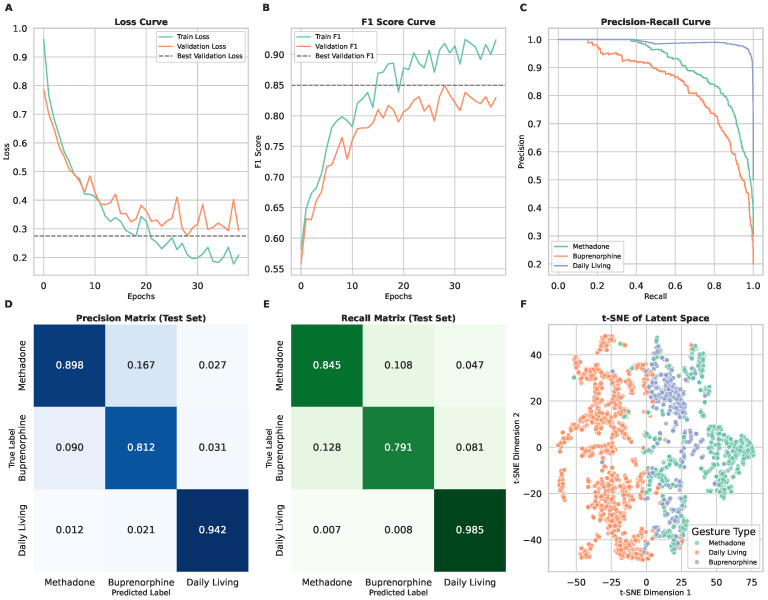
Model performance and gesture classification results for Scenario 2. In this scenario, the model classifies gestures into three categories: methadone, buprenorphine, and daily living. (**A**) Training and validation loss curves demonstrate model convergence over epochs. (**B**) F1 score curves for training and validation sets indicate classification performance across classes. (**C**) Precision–recall curves highlight the model’s ability to distinguish between the three gesture categories. (**D**) Precision matrix of the test set, showing the proportion of correct predictions for each class. (**E**) Recall matrix of the test set, illustrating the sensitivity of the model to each class. (**F**) t-SNE visualization of latent gesture representations, demonstrating the separation between the methadone, buprenorphine, and daily living gestures.

**Figure 6 sensors-25-02443-f006:**
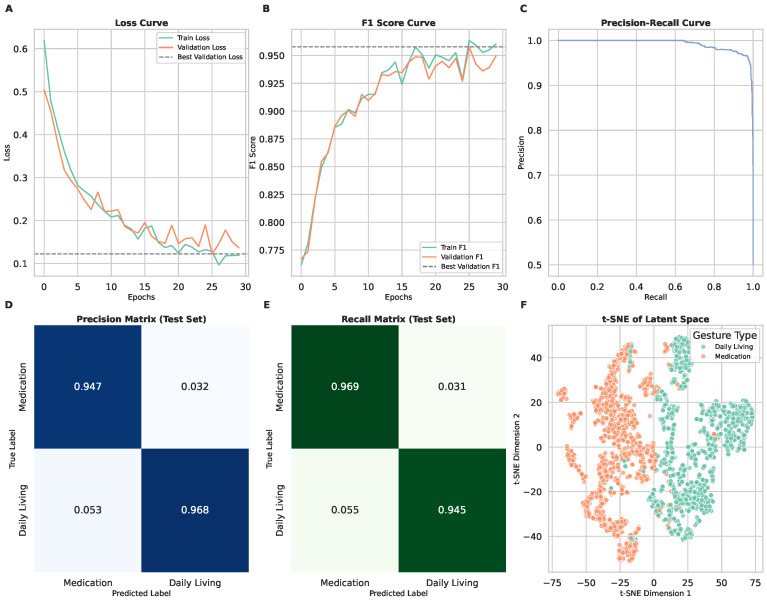
Model performance and gesture classification results for Scenario 3. In this scenario, the model performs binary classification to distinguish between medication-taking gestures and daily living activities. (**A**) Training and validation loss curves showing model convergence and generalization over epochs. (**B**) F1 score curves for training and validation sets, representing classification performance. (**C**) Precision–recall curve illustrating the trade-off between recall and precision for detecting medication-taking gestures. (**D**) Precision matrix of the test set, showing the proportion of correct predictions for each class. (**E**) Recall matrix of the test set, highlighting model sensitivity to medication-taking gestures. (**F**) t-SNE visualization of latent space representations, demonstrating separation between medication-taking and daily living gestures.

**Table 1 sensors-25-02443-t001:** Performance metrics across different scenarios.

	Scenario 1	Scenario 2	Scenario 3
Macro Performance			
F1 Score	0.89	0.88	0.96
Recall	0.88	0.87	0.96
Precision	0.89	0.88	0.96
Methadone Performance			
F1 Score	0.91	0.87	0.96
Recall	0.92	0.85	0.97
Precision	0.90	0.90	0.95
Buprenorphine Performance			
F1 Score	0.86	0.80	0.96
Recall	0.84	0.79	0.97
Precision	0.88	0.81	0.95
Daily Living Performance			
F1 Score	NA	0.96	0.96
Recall	NA	0.99	0.94
Precision	NA	0.94	0.97

## Data Availability

The data presented in this study are available on request from the corresponding author.

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
