# Peer review of "Monitoring Opioid-Use-Disorder Treatment Adherence Using Smartwatch Gesture Recognition"

_sensors, 2025, doi:10.3390/s25082443_

Round 1
Reviewer 1 Report
Comments and Suggestions for Authors
The paper explores the feasibility of using machine learning algorithms and consumer-grade smartwatches to identify medication-taking gestures for opioid use disorder treatment. The study focuses on methadone and buprenorphine adherence through gesture recognition. Data was collected from 16 participants who simulated medication-taking gestures over two weeks. A RegNet-style 1D ResNet model was employed for classification, achieving high F1 scores across three scenarios: distinguishing medication types, differentiating medication gestures from daily activities, and detecting medication-taking gestures. The study demonstrates the potential of smartwatch-based monitoring for improving treatment adherence in OUD management.
Drawbacks:
1. The study includes only 16 participants, predominantly young females.
2. The study acknowledges that methadone and water ingestion gestures are similar, potentially leading to misclassification.
3. While the RegNet-style 1D ResNet is effective, the study does not compare its performance with other deep learning architectures.
Recommendations:
1. Expanding the sample size to include a more diverse demographic to assess performance variations across different populations will increase the quality of the study.
2. Explore multi-modal data integration (e.g., heart rate, time-of-day patterns) to improve gesture differentiation.
3. To assess relative performance, benchmark the model against other architectures, such as LSTMs or Transformer-based models.
4. Implement SHAP or Grad-CAM to analyze model decisions and improve transparency.
Reviewer 2 Report
Comments and Suggestions for Authors
Your paper innovatively applies smartwatches and machine learning to OUD treatment medication adherence monitoring, providing a new solution for patient management. It demonstrates the potential of deep learning models in identifying and classifying medication-taking gestures, laying the foundation for the development of more precise and personalized interventions in the future.
The topic is interesting, but there are some major concerns that that need to be addressed:
1、The study data were collected from a simulated environment, not real-world medication intake behavior, which may lead to a decrease in model performance in practical applications. It is recommended to conduct clinical validation in future research to assess the effectiveness of the model in real-world scenarios.
2、The study sample primarily consists of young women, lacking representation of other age, gender, and racial groups. It is recommended to expand the sample range to improve the generalizability of the research findings.
3、Due to the similarity between medication-taking gestures and some daily activities, the model may produce false positives and negatives. It is recommended to explore the integration of more sensor data and methods, such as combining heart rate, blood oxygen, and other physiological indicators, to improve the model’s discrimination ability.
4、There are individual differences in gestures, and the model needs to adapt to this variation. It is recommended to develop personalized machine learning models to improve the robustness and accuracy of the model.
5、It is recommended to cite the following articles to improve the content of the paper and make it more comprehensive:
①Hu F, Zhang L, Yang X, et al. EEG-Based Driver Fatigue Detection Using Spatio-Temporal Fusion Network With Brain Region Partitioning Strategy[J]. IEEE Transactions on Intelligent Transportation Systems, 2024, 25(8): 9618-9630.
②A. -J. Long and P. Chang, "The Use of Health Smart Card in Bridging Discontinuity of Care for Pregnant Woman," 2009 Sixth International Conference on Information Technology: New Generations, Las Vegas, NV, USA, 2009, pp. 1492-1497, doi: 10.1109/ITNG.2009.301.
③M. Aldeer et al., "MedBuds: In-Ear Inertial Medication Taking Detection Using Smart Wireless Earbuds," 2022 2nd International Workshop on Cyber-Physical-Human System Design and Implementation (CPHS), Milan, Italy, 2022, pp. 19-23, doi: 10.1109/CPHS56133.2022.9804515.
Comments on the Quality of English Language1、Some sentences are too complex and can be split into shorter ones for better readability. For example, in the first paragraph, the sentence “This study explores the feasibility of using machine learning algorithms alongside consumer-grade smartwatches to identify medication-taking gestures for OUD treatment, specifically focusing on methadone and buprenorphine.” can be split into two sentences.
2、Some word choices are not precise enough and can be replaced with more appropriate words. For example, in the second paragraph, “These gestures, while different in intent, share similar movement patterns, making them challenging to differentiate.” can be revised to “These gestures, while differing in intent, share similar movement patterns, making them difficult to distinguish.”
Reviewer 3 Report
Comments and Suggestions for Authors
I have reviewed the manuscript titled "Monitoring Opioid Use Disorder Treatment Adherence Using Smartwatch Gesture Recognition" and find it to be a well-structured and promising study that demonstrates the feasibility of using smartwatch-based gesture recognition and machine learning to monitor medication adherence in opioid use disorder (OUD) treatment, particularly for pregnant women.
The manuscript presents a compelling study on the use of smartwatch gesture recognition and machine learning to monitor medication adherence in opioid use disorder (OUD) treatment, particularly for pregnant women. The research is well-structured, and the results are promising, demonstrating the feasibility of using consumer-grade smartwatches to detect medication-taking gestures with high precision and recall. However, several revisions could enhance the clarity and overall impact of the study.
Abstract
1. The abstract could briefly acknowledge the study's limitations, such as the use of simulated gestures and the homogeneous participant pool.
Introduction
1. While the context of the opioid epidemic is well-presented, the transition to the technological gap being addressed could be smoother. Emphasising the limitations of current adherence monitoring methods and how smartwatch-based solutions could address these challenges would strengthen the rationale for the study.
Methods
1. The description of the data collection process is thorough, but a visual aid, such as a flowchart, could help illustrate the steps involved in the natural and scripted medication-taking event (MTE) phases.
2. While the use of simulated gestures (e.g., water and Listerine strips) is practical for initial feasibility studies, the manuscript should explicitly state this as a limitation, as real-world medication-taking gestures may differ.
3. The discussion of the RegNet-style 1D ResNet model is clear, but a more detailed justification for choosing this architecture over others, such as LSTMs or Transformers, would strengthen the methodological rigor.
Results
1. The manuscript should explore why the model performs slightly better for Methadone gestures compared to Buprenorphine.
2. The t-SNE plots are useful but could benefit from a more detailed interpretation, particularly regarding the similarity between Methadone and Buprenorphine gestures and any potential outliers.
3. The high performance in Scenario 3 (binary classification of medication-taking vs. daily living) is impressive, but the manuscript should discuss whether this performance is sufficient for real-world deployment, especially given the potential consequences of false positives or negatives in clinical settings.
Discussion
1. How does the impact of different smartwatch models, variations in how participants wear the watch, and the potential for overfitting due to the relatively small dataset should be addressed
2. The manuscript should discuss how the model could be integrated into existing healthcare systems and the ethical considerations of continuous monitoring, particularly for pregnant women.
By addressing these points, the manuscript will be more robust, clearer, and better positioned for publication. I recommend a minor revision to address this points
